# The Design of a Safe Charging System Based on PKS Architecture

Jianhong Zeng [1], Yi Zhang [1], Youhua Xue [1], Wenqi Li [1], Jing Li [1], Linchao Zhang [2] and Shipu Zheng [1,*]

1    The Key Laboratory of PK System Technologies Research of Hainan, Chengmai 571900, China
2    Department of Computer Engineering, Jeju National University, Jeju 63243, Korea
*    Correspondence: zhengshipu@jiri.ac.cn; Tel.: +86-898-3286-7419

**Abstract:** With the development of new energy vehicles, information sharing and charging service-sharing in the Internet of Vehicles have become popular directions in smart city research. The number of new energy vehicles has surged, and the ensuing range anxiety and low charging efficiency have become the main problems to be solved urgently in charging services. In the era of big data, privacy leakage is becoming more and more serious, and information security and privacy protection cannot be delayed. This paper proposes an efficient charging and privacy protection system based on the PKS system. The original single-stage topology is improved by adding the PFC and LLC circuit topologies. The PID method is used to precisely control the voltage and current loss to improve the conversion efficiency of the charging pile. The private data in the shared information uses the RSA encryption algorithm to prevent the leakage of private data and enhance the security of system communication. This paper aims to improve the charging efficiency of charging piles and the security of private information in network communication. Simulation experiments are carried out on the proposed hardware topology and software encryption system scheme. Experiments compare the waveform state of the improved output current and voltage and the safety protection area of the system architecture. The results show that the proposed charging system is efficient and safe.

**Keywords:** Phytium–CPU Kylin–operating Security (PKS) system; Rivest Shamir Adleman (RSA) algorithm; new energy vehicle; charging pile; control system



## 1. Introduction

Due to automobile exhaust emissions, environmental pollution is becoming more serious. New energy vehicles are becoming the best mode of transportation, characterized by no exhaust gas and clean energy for driving. Presently, the related technologies of electric vehicles are well-developed. However, the charging problem must be solved for new energy vehicles. At present, charging piles have the following issues. First, there are few charging piles; second, the charging time is long, and the charging efficiency is not high. Third, there is no sound information management platform for reciprocal links and integration [1], which leads to the inability to conduct deep data processing, performance detection, and the working status of the charging pile energy system, and has a serious impact on the security of charging pile user data and user experience [2].

Compared with developed countries such as the United States, the development of new energy vehicles in China started later, but the development momentum in recent years cannot be underestimated. With the further increase of the state's subsidies and other policies in the field of new energy vehicles and the improvement of people's awareness of environmental protection, people increasingly choose new energy vehicles as purchasing objects [3]. The Notice on Improving the Financial Subsidy Policy for the Promotion and Application of New Energy Vehicles shows [4] that the financial subsidy policy for the promotion and application of new energy vehicles will be extended to the end of 2022. In addition to the unified national policy, various localities have successfully introduced a

series of corresponding incentive measures to promote the development of new energy vehicles. At the same time, although the current new energy vehicle and battery energy technology are constantly developing, the layout of charging piles is still not mature enough. The ratio of domestic new energy vehicles to new energy vehicle charging piles is only 3.4:1, of which the ratio to public charging piles is only 7.3:1. The ratio of the vehicles to charging piles fails to support the development of China's new energy vehicles. Infrastructure is still a shortcoming that restricts the development of China's new energy vehicle industry [5].

Moreover, the charging pile is difficult to set up in a residential area. The charging takes a long time, the installation position is unreasonable, and availability is unbalanced in terms of region. Network security and other problems also seriously affect the quality of the green development of China's new energy automobile industry.

This paper studies the PKS System and the information platform framework of new energy vehicles based on this system. This paper introduces the components and application functions of the PKS System at all levels. The technical innovation of the PKS System studies the current charging pile information platform and compares the encryption algorithms of privacy protection. It concludes that the environment layer needs to solve the problem of elastic resource allocation of big data platforms to meet the different requirements of different components or applications on hardware resource performance. At the same time, the monitoring requirements of hardware [6], cloud platform, and system components are provided. Only in this way can we genuinely start from business, improve the ability to monitor efficiency and quality, unify the interface of cloudy resources, support the unified arrangement of gray resources, and serve the unified management of heterogeneous data centers [7]. Table 1 lists the new energy indicators in new energy charging.

**Table 1.** Electrical Performance Index of New Energy Vehicle Charging Pile.

| Input: Three-Phase 380 VAC + 15% | Input AC Power Frequency: 50 HZ 10% |
| --- | --- |
| Output power: 0–12 KW | Output voltage: 20 V–75 V |
| Steady current accuracy: 2% | Voltage stabilization accuracy: 1% |
| Maximum power: >90% | Power factor: >0.9 |

Section 1 introduces the new energy charging piles' low efficiency and privacy and a proposed efficient and safe charging system for new energy vehicles based on the PKS System. Section 2 presents some solution systems for new energy charging piles and constructs the security information platform at the software and privacy levels. Section 3 describes the security of the charging system based on the PKS System, encrypts the communication process with the charging pile by the RSA encryption algorithm to protect privacy, and makes a security analysis. Section 4 describes the overall design scheme of the automobile charging pile control system, designs the structure of the current-voltage controller and SVPWM mode of the active power filter system, and improves the charging efficiency and low energy consumption of the charging pile system. In Section 5, the proposed charging system is tested, and the protection ability and security of the PKS charging encryption system are compared. The topology network of the new energy vehicle charging pile is simulated, and the results are in line with the proposed system. Section 6 summarizes the proposed system scheme and explains the related work to be studied in the future.

## 2. Related Work

This paper studies the PKS System and the information platform framework of new energy vehicles based on this system. This paper introduces the components and application functions of the PKS System at all levels. The technical innovation of the PKS System studies the current charging pile [8] information platform, compares the encryption algorithms of privacy protection and concludes that the environment layer needs to solve the problem of elastic resource allocation of the big data platform to meet the different requirements of different components or applications on hardware resource performance. At the same

time, the monitoring requirements of hardware, cloud platforms, and system components are provided. This way, we can start from a business perspective, improve the ability to monitor efficiency and quality, unify the interface of cloud resources, support the unified arrangement of cloud resources, and serve the unified management of heterogeneous data centers. We also contend that establishing a safe and reliable information platform for the charging piles of new energy vehicles is feasible.

### 2.1. Related Research on the PKS System

The security foundation of the PKS System is based on the Phytium chip and Kylin operating system, the built-in trusted Computing 3.0 technology, and a converged system of "trusted computing + conventional computing". PKS meets the requirements of national security standards and policies, integrates a cryptographic system, authentication system, and trusted computing 3.0, and standardizes conventional attack and defense tools. It can realize software-defined physical isolation [9] for computing, memory, I/O, network, storage, and other components, as well as dynamic monitoring of the running process, and realize all elements and life cycle of intrinsic security + process security. It is suitable for intranet, extranet, internet, and mobile environments.

As shown in Figure 1, the PKS System is divided into three levels and eight parts, with the following functions:

1.  The primary layer (the lower two Trusted 3.0, which integrates the network attack and defense foundation is built with a Feiteng CPU, Kirin OS, Great Wall motherboard, and BIOS, protects physical security memory, provides infrastructure services, and realizes security functions such as physical isolation, password, authentication, trustworthiness, and auditing).
2.  The system security layer, that is, a safe operation framework, integrates system ecology, industry alliance, and application support and realizes the secure operation functions of the system, such as operation environment, network, cloud resources, and data services.
3.  The security management layer (on the right side and the top layer), including the analysis and protection, back-end management center and other two parts, uses big data + AI bypass analysis and monitoring to achieve security strategy, attack and defense tools, security management, security control and other process security management and control functions.

The PKS architecture technology builds an integrated new software and hardware compatible system from the bottom. The PKS architecture has 3 technical innovations:

*   Built-in trusted CPU technology [10]
*   First time in the world. Feiteng CPU isolates the trusted core for trusted computing and embeds the safelist mechanism into Kirin OS to realize the dual computing and security protection system. Second bullet;
*   For the first time in the world, the built-in physical protection technology in memory is adopted. A hardware protection chip is built into the memory cache controller, and the key codes and data can be managed in real-time according to requirements;
*   Unified terminal security center and cloud unified security management and control. A unified terminal security protection software portal and cloud unified security control interface integrate conventional security software with PK bottom core, improving security protection effects and efficiency.

### 2.2. Charging Pile Network Information Platform

The construction of the network platform is to establish the information platform of the new energy vehicle charging pile based on the PKS System. The management platform of vehicle user data and the big data information platform are considered first.

*   The big data environment is built on the IaaS layer of the cloud platform, including the hardware resource layer, basic software layer, and virtual resource layer.

- The hardware resource layer provides essential hardware resources such as servers, storage devices, and network devices for big data platforms.
- The basic software layer builds on the system environment of PKS architecture with the Kirin operating system and independent security middleware software.
- The virtual resource layer is based on an OpenStack open-source framework, builds the virtualization platform of IaaS, realizes the unified management of computing, storage, network, and other resources, and provides an efficient resource allocation and management mechanism for the big data platform.

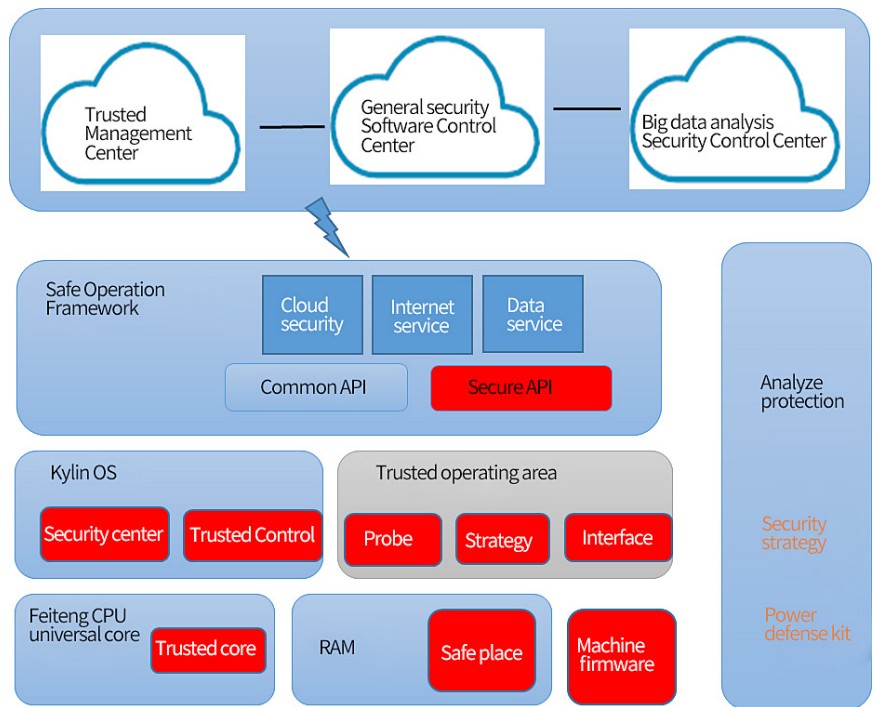

**Figure 1.** Overall system diagram.

Charging pile access technology: Data access technology can provide rich data access interfaces, including supporting multi-source data access methods [11], such as files, databases, streaming data (video, Internet of Things data); Internet data access is supported, and relevant data information can be obtained through interactive information flow. At the same time, real-time data collection can be realized by car–machine interaction, as shown in Figures 2 and 3.

- The data source layer is mainly responsible for collecting vehicle data. The collection method can be carried out through the remote monitoring system of new energy vehicles. At the same time, the data during vehicle maintenance is also used as an essential data source for the data source layer.
- The data storage layer classifies and stores the data according to the different characteristics of the data, such as the parameters of crucial parts, vehicle parameters, and user parameters. The cloud service platform of the charging pile combines the data according to their respective properties through the data warehouse and market management, and the supply support layer processes the data.
- The application support layer is the core of the data management platform. It carries out relevant statistical work such as diagnosis and data and service reports based on the data provided by data sources and engineering experience.
- The users access the user interaction layer through standardized interfaces, including engineering developers, sales, after-sales personnel, and users' applications.

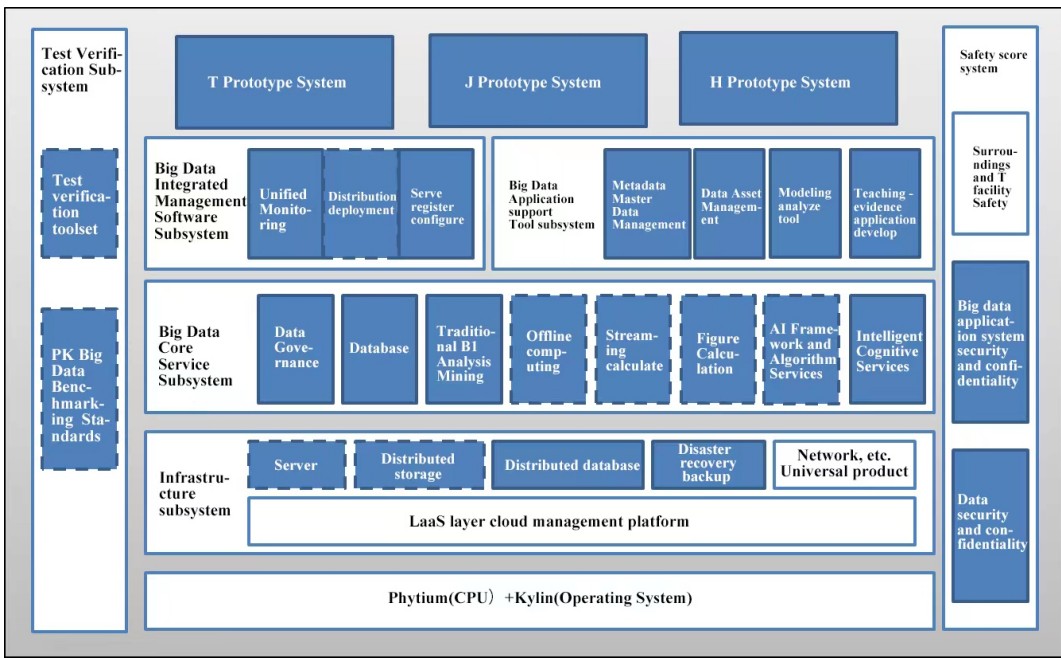

**Figure 2.** Big Data information platform diagram of the PKS system.

**PKS system data management platform**

Mobile terminal

Large screen display

Internet

4G Base station

USR-G806

Charging pile

Charging pile

Charging pile

**Figure 3.** Big Data application support technology platform diagram of the PKS system.

*2.3. Research on Safety Protection Status of New Energy Charging System*

As the technology of energy conversion of charging piles to new energy charging piles develops, the problem of network information security of the charging banks has attracted the attention of domestic and foreign scholars. The continuous formation of the industrial ecological chain of electric vehicles will attract more attention.

According to the analysis of the network information security requirements of charging pile networks in various countries, work has been done to set the relevant electrical standards for banks, the communication standards for ISO-TCP/IP [12] evaluation facilities, and the use of RSA encryption algorithms. The literature [13] describes the distribution of charging pile resources encountered in the green and rapid development of new energy vehicles, the communication security problems between users and charging piles, and the data analysis process. According to the characteristics of new energy vehicle charging pile data in the massive data scenario, it can provide more practical knowledge, more accurate information and more timely responses for new energy vehicle charging pile operators and charging users to the maximum extent. The utilization rate of charging piles should be improved, and the operators should have assistance in adjusting the operating strategy to provide data support.

The literature [14] proposes a two-way large change ratio DC–DC converter based on the photovoltaic charging pile of new energy vehicles. The voltage change ratio is improved by applying coupling inductance technology, and the bidirectional change and electric energy processing are realized. Another study [15] elaborated on the construction and optimization analysis of charging piles for new energy vehicles and made reasonable solutions to the needle problem. Other research [16] provided an understanding of how to generate RSA key pairs (public key and private key) through the logical derivation and mathematical derivation processes of the RSA encryption algorithm, which meets the security and privacy requirements of information flow between the client and the charging pile. The research in [17] describes the overall hardware frame design of the charging bank for new energy vehicles, and the chip STC89C52 is used as the core controller. The inductance is held by controlling the effective area in the turn structure, and the inductance effect of the charging pile is treated equivalently. The charging loss of the charging system for new energy vehicles is reduced.

The new energy charging system is in the early stage of development. The potential safety hazards in charging facilities and information security regarding setting hardware topology and physical security need to be analyzed significantly. Solve the safety problems existing in the charging system of electric vehicles. At the same time, the charging system's high efficiency and low energy consumption are also areas of grave concern. It is also an urgent problem in the new energy electric vehicles field.

## 3. Privacy and Security of Charging Information System

We propose a new energy charging system based on the PKS System. The new energy charging system consists of the client application, encrypted transmission, authentication and verification, server system, charging pile system, power grid control system, and car and machine system. This paper focuses on the research of client encryption transmission and charging pile systems. This section describes the system hierarchically. The network is a connection-oriented, reliable, byte-stream-based secure communication link based on ISO-TCP/IP protocol under an encryption system. Before the connection between new energy vehicles and the charging piles of the new energy vehicles is established, their legality can be verified by mobile phone identity authentication or IC card identity authentication in two ways, guaranteeing the legitimate transactions of both communication parties. After verification, both parties can establish a reliable, secure, and safe link for data exchange encrypted by an encryption system. This protocol completes the highly confidential data exchange between customer information flow data and power flow data. User identity authentication of mobile phones and IC card can get personalized service of new energy vehicle charging piles.

Figure 4 shows the user calling the application function interface through the client, the car and machine control, remote charging, and information inquiry services applied in this paper. The information requested by the client is encrypted by the information encryption system and then transmitted to the server through multiple transmission channels, which involve roadside units and intermediate routes in the system—the security of the middleware pile, new energy vehicle, and the safety of information flow. The PKS system is deployed in the physical hardware to support the operation of the whole network.

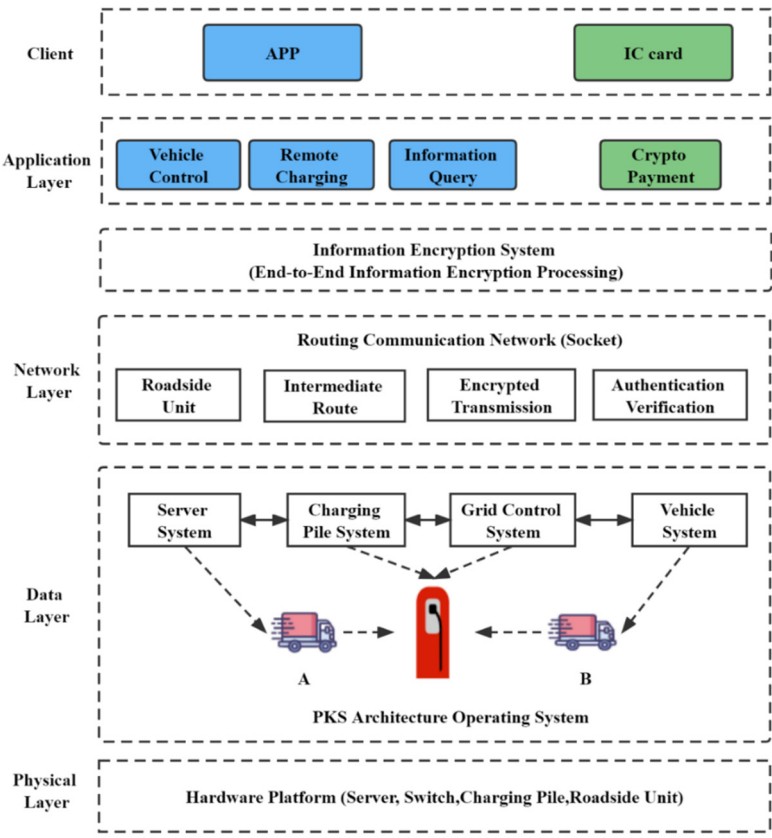

**Figure 4.** System diagram of new energy charging system based on the PKS system.

At the level of customer information flow, customers have the functions of controlling, remotely charging, and displaying information on the internal battery of the new energy vehicle and the user's information operation interface through their mobile phones. How to make the remote operation of customers more reliable? The secret information sent by mobile phones is encrypted, and the RSA encryption algorithm, which encrypts short and small data keys and passwords, is not too long and has good security. RSA's public and private keys protect the sent information double. The network layer socket [16] is an abstract layer through which applications can send or receive data and open, read, write and close it like files. In the PKS system network, the unique encryption ID is set for the charging pile to ensure the uniqueness of the charging bank. The encrypted information of the user end is decrypted and sent to the charging pile to complete remote charging after being processed by PKS operating system. The operation process of an IC card is similar to that of mobile phone transmission, except that after swiping the card, an encrypted payment process identical to bank password input is required.

The power flow level of the vehicle system, the battery loss inside the vehicle, the loss of components, the output efficiency of the electric vehicle motor, the failure of the internal components of the new energy vehicle charging pile, the charging efficiency of new energy vehicle charging a fortune, the charging amount of users, and the privacy information of users are not only a significant challenge to the network information flow

but also a challenge to the compatibility between the operating system based on the PKS framework and new energy vehicle charging pile. Here, we made two significant optimizations. Through the hardware level—the scheme of changing the circuit topology—the PFC and LLC shift the flow of closed-loop feedback control of DC-AC-DC plus PI control algorithm for its output current, voltage and efficiency. It was found that the THD harmonic [18] is less than the controllable range, and the electrical index performance of the charging pile of a new energy vehicle can also reach the standard. However, this is only an experimental process. As such, we are considering using relatively stable and capable detecting and dispatching power parameters to replace the electrical indicators that do not need topology innovation and classical control algorithms to meet our needs. The charging pile of new energy vehicles based on the PKS System can better dispatch the collected power parameters (voltage, current and electrical efficiency) in the operating system of the PKS system, and the charging pile system can accurately adjust the power parameters and achieve a relatively stable and safe charging process with the vehicle and machine system.

After charging, the lost power will be converted into the remaining amount according to the server system in the data layer. Because there is a risk of privacy data leakage when it is transmitted to the user's mobile phone, it goes through the network layer, and the user's unique ID is matched through authentication. The double encrypted transmission of the encryption system RSA + TEA is decrypted by the customer's mobile phone to display the user's information, electricity consumption and the remaining amount.

*3.1. RSA Encryption Algorithm*

As shown in Figure 5, RSA algorithm derivation formula flow:

- Choose a pair of different and large enough prime numbers $p$, $q$.
- Calculate $n = pq$.
- Calculate $f(n) = (p-1)(q-1)$, and keep $p$ and $q$ strictly confidential so that no one knows.
- Find a number e which is coprime with f(n), and $1 < e < f(n)$.
- Calculate d so that $d \equiv 1\ mod\ f(n)$. This formula can also.
- To explain, here $\equiv$ is a symbol for unity in number theory. In the formula, the left side of $\equiv$ the sign must be congruent with the right side of the movement. That is, the modulo operation results on both sides are the same. No matter what value $f(n)$ takes, the development of 1 mod $f(n)$ to the right of the symbol is equal to 1; the product of d and e on the left side of the mark must also be similar to 1 after the modular operation. It is necessary to calculate the value of d so that this congruence equation can hold [19].
- The public key $KU = (e, n)$ and the private key $KR = (d, n)$.
- When encrypting, the plaintext is first converted into an integer M from 0 to $n-1$. If the plaintext is long, it can be divided into appropriate groups and then exchanged. If the cipher text is $C$, the encryption process is Formula (1):

$$C = M^d (\text{mod n}) \tag{1}$$

- The decryption process is as Formula (2):

$$M = C^d (\text{mod n}) \tag{2}$$

The secret to the RSA algorithm's secrecy lies in the private key's security. The public key (n, e) is shared. If an attacker wants to obtain D, he must bring two prime factors, the P and Q of n. For one, the decomposition factor is challenging in mathematics and engineering, and the proof of the complementarity between public key E and private key D of the RSA algorithm needs to be proved by the Euler theorem [9].

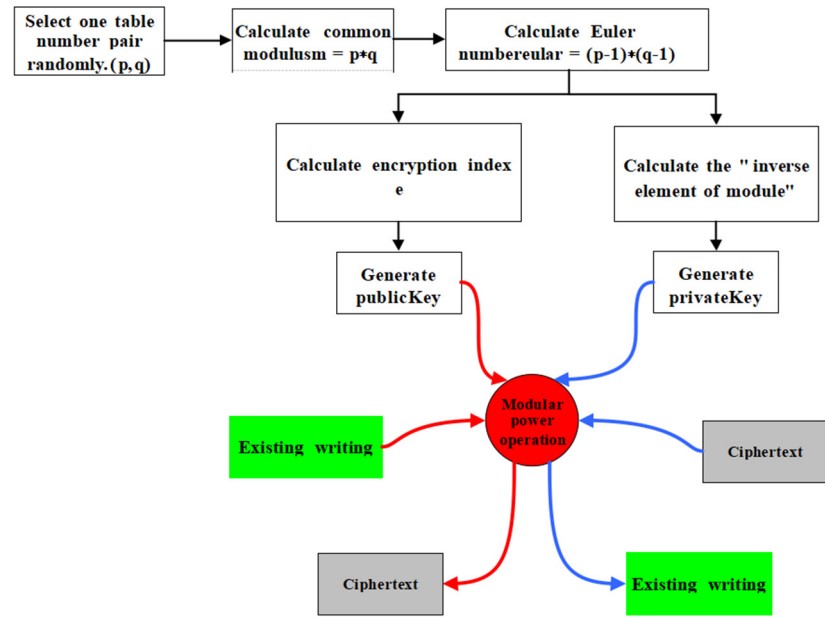

**Figure 5.** Deduction flow chart of RSA algorithm encryption process.

### 3.2. Network Encryption and Decryption Process Based on the PKS System

Based on the introduction of the overall system of the PKS charging pile for new energy vehicles, the interaction process between the client and the charging bank for new energy vehicles is mainly introduced based on client information flow and electric flow. In the interactive process, an RSA-based encryption algorithm will encrypt the information between the information flow and the electric flow. Then, the key will be distributed and managed to achieve double protection [20].

As shown in Figure 6, this section analyzes the customer information and the internal electrical structure of the new energy vehicle charging pile based on the RSA encryption algorithm. The following is a detailed introduction to the process of encryption and decryption between the client and the new energy vehicle charging pile by the encryption algorithm.

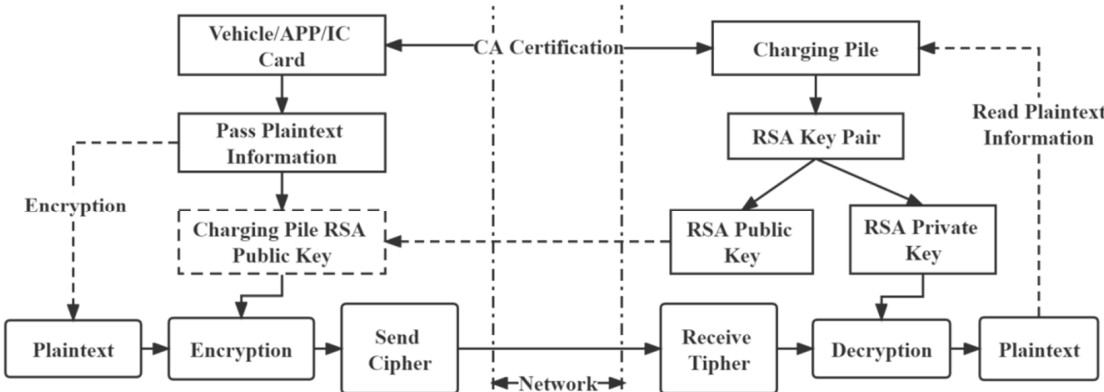

**Figure 6.** Flow chart of encryption and decryption process between a user and charging pile.

The client information flow mainly transmits the user's basic information, the remote charging instruction of the charging pile, the information about the internal battery of the charging rise and the function of the customer's information operation interface through the car machine, APP and IC card as the plaintext information instruction M, and the sending ciphertext C is formed by the RSA encryption algorithm. The RSA public key (*e*,

*N*) of the charging pile is included in the encryption process. The encryption formula is as Formula (3):

$$m^e \ mode \ N = c \tag{3}$$

The sent key is distributed to the service system of the charging pile through PKS operating system for internal decryption. As the RSA algorithm comes with a private key for decryption, the ciphertext C needs to be accepted in the decryption process. In the decryption process, selecting the appropriate E and N is necessary, and there must be a D, which constitutes RSA private key pair (d, *N*) to decode the ciphertext C to obtain the plaintext M. The decryption formula is as Formula (4):

$$m^e \ mode \ N = m \tag{4}$$

Through the process of transportation encryption and decryption, it is clear that the RSA encryption algorithm is safe and reliable. The encryption and decryption keys are separated, making the keys' distribution more convenient. They can significantly protect the personal privacy information of users, the current distribution inside the charging pile of new energy vehicles and the network security of the power grid. The recent electric analysis is as above.

As shown in Figure 7, the main task of CA certification is to accept applications for digital certificates, issue digital certificates and manage digital certificates. As a trusted third-party organization, the CA bundles the user's key with other identification information and assumes responsibility for the validity of the public key in the public key system. The CA certification center is responsible for the application, issuance, production, and verification of digital certificates. Revocation, authentication, and management provide network customer identity authentication, digital signature, electronic notarization, and secure e-mail. As shown in Figure 6, the user's APP/IC card and the charging pile establish a CA certification form. The user's private information and charging balance information are bundled with the charging pile system.

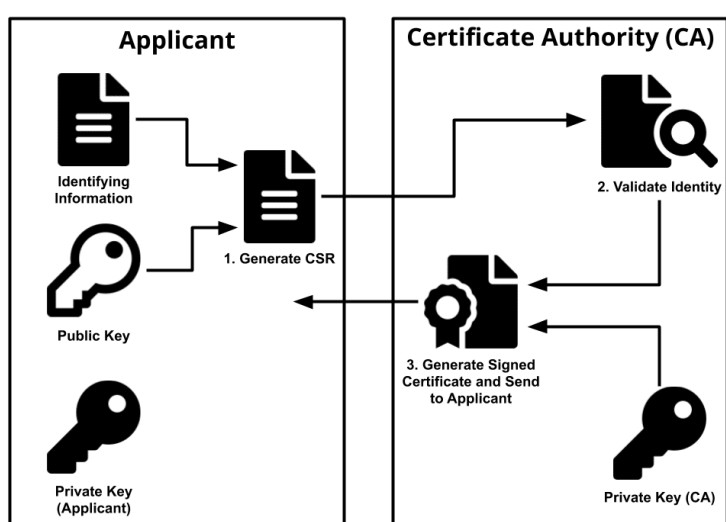

**Figure 7.** The process of CA certification.

### 3.3. Interactive Protection of Charging Pile Privacy Information

The privacy information of new energy vehicle charging pile users based on the PKS System includes not only identity information, living habits, new energy vehicle location and other private information, including closely related information such as battery usage status, internal faults of new energy vehicle charging pile, charging and discharging efficiency of charging pile, running track of the electric vehicle, but also the charging method of new energy vehicle charging pile and the transaction amount flow in the server.

Through the connection of the charging bank, it is possible to read the trajectory of the electric vehicle's internal BMS manager, adjust the user's sitting position on the touch screen and the handle of the steering wheel of the electric car, and read the customer's living habits and identity information. Therefore, it is a significant problem to be solved in the current network information security of the charging pile of new energy vehicles to form the privacy content of the information network among the charging rise of new energy vehicles, the server and the users.

- Privacy [21] protects the factual identity information [22] of charging pile users, including the user's real name, actual address, home address and ID number. In setting the pile system of new energy vehicles based on the PKS System, personalized verification methods such as face recognition and digital signature are added to authenticate the real identity of users.
- Data privacy refers to the server's data storage, such as the location, home address, charging service request and other charging service contents of users, by the convergence unit and setting pile background service center to provide better service. Usually, network intruders often attack the storage devices inside the server, which threatens the security of charging pile users in the process of using charging piles.
- In the era of intelligent charging, the connection between new energy vehicles and charging piles will involve the location privacy and battery life privacy of charging cars. Suppose the user's location privacy is leaked. In that case, it will be provided to intruders to track and monitor the user's habitual movement trajectory and location direction and seize the corresponding laws of users to steal personal life trajectory and privacy information. On the other hand, the privacy of battery life In the process of tram charging, intruders can also attack the server and car system by implanting hardware into the charging pile to implant control viruses and operate the vehicle to endanger driving safety.
- The privacy of the charging pile's electric charging. According to the law of developing a new energy industry, China's development is the most robust. The connection of the new energy grid is very complicated and operates efficiently, and the charging pile network controlled by intruders endangers the safety of the charging pile cluster. At the same time, stealing the charging pile's hardware information affects the electrical network's security based on the electrical reference index, so the safety of the electrical index of the charging pile itself is also the focus of our research.

### 3.4. Privacy Security Based on the PKS System

### 3.4.1. PKS Protection Principle

First, in the past, trusted computing technology relied on the motherboard to isolate I/O, which not only made the motherboard not universal but also created its high cost, and the bus could not be separated after finally converging in the CPU. The PKS has built-in trusted computing technology, which can be physically isolated by software. The original software does not need to be replaced, so it has high compatibility and little change. This scheme is safer, lower in cost, more efficient operation and more flexible in management. Secondly, in the past, trusted computing caused some disadvantages, such as difficulty installing user software, poor user experience, and minor update of the virus database and patch package, due to the trusted safelist mechanism. The PKS integrates the safelist tool, password system, protection terminal and OS to realize the automatic authentication "white-brushing mechanism" of software stores and terminal safelists, which is truly safe, efficient and easy to use. Third, PKS is added to secure memory control, and the memory physical protection mechanism of the vital data area and key code area is constructed, which plays an essential role in security protection against vulnerability attack, vulnerability penetration and virus spread. Fourthly, PKS is an integrated design and system from the bottom chip to the upper application. At the same time, it standardizes the security interface standards so that all security control points can be connected and

cooperate tacitly and controlled and processed by a unified management and control center, with better protection and more efficient management.

The essential protection of PKS includes physical isolation, a password system, authentication, trustworthiness, auditing, and monitoring as Figure 8.

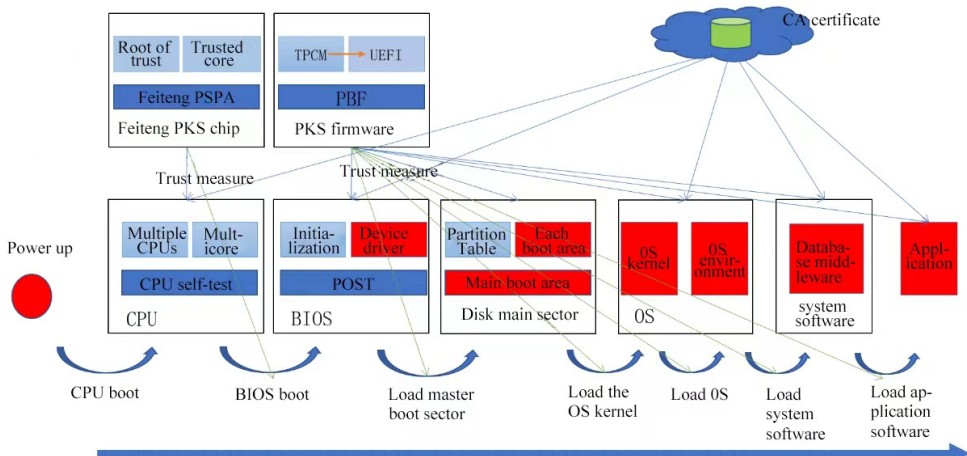

**Figure 8.** Computer startup and software running security protection process.

From the computer's start to the software's operation, PKS has security measures in every link. The primary means is to take the trusted root established based on cryptography and physical protection as the reliable source.

### 3.4.2. Operating Efficiency of PKS

The impact of loading a PKS safety protection system on the whole machine's performance is about 10%, far less than the traditional safety protection measures. The function, stability and compatibility of the entire PK system after loading PKS have not changed much. Among them, after the PKS transformation of the PK server, benchmark tests and application tests such as Spec2006, UnixBench, IOZone, and SpecJVM2008 show that the server's overall performance drops below 10%, and the cost of security resources meets expectations. For the FT-2000/4 new quad-core terminal, because one core is assigned to take the role of TPCM, the computing power of the whole machine is borne by the remaining three cores, and the overall performance loss is 25%. However, most applications on the terminal, such as the browser and word processing, will not exceed four processes. Presently, applications in the airport rarely need to use more than four independent computing cores simultaneously, so the overall performance loss of the terminal can be neglected in practical applications.

### 3.4.3. PKS Protection Efficiency

Under the protection of PKS, standard network attack methods cannot control the attacked computer. Some of these include:

- Anti-network attack and vulnerability penetration, which can prevent network attack and vulnerability penetration;
- Anti-Trojan and viruses—It can effectively protect all virus Trojans that attack the CPU, firmware, boot area, OS and kernel, application environment, etc.;
- Data protection—Can prevent ransomware from encrypting and destroying files, memory data attacks, database SQL injection, AI data pool vulnerability, storage server Trojan horse and viruses, and network transmission data theft;
- Anti-proliferation-Anti-Trojan viruses spreading through the network;
- Anti-performance degradation—Can avoid system performance degradation caused by loading conventional security software and virus infection.

- Slow emergency response prevention, automatically update patches, automatic AI disposal and situation analysis of security risks, automatically update policies, and collect real-time status of the whole system around the clock.
- Physical attack prevention—Can prevent hardware and side-channel attacks, voltage frequency attacks and hardware vulnerability attacks;
- Prevent risks in the process of software development and software release;
- Content prevention—Can prevent the use of non-compliant, substandard and unqualified software, the management of classified non-compliant documents and the printing and burning of non-compliant documents;
- Prevent the AppStore from being replayed.

### 3.4.4. Security Analysis

Combined with the RSA encryption algorithm and the internal mechanism of PKS, the whole new energy vehicle charging pile is analyzed. For the data control center of the PKS-based new energy vehicle charging pile system, the background management center based on PKS can only accept the anonymous identity information of the charging pile users and the status information of new energy vehicles. For the server system of the PKS operating system, the connected electric vehicle users are legally CA-certified [23] and RSA-encrypted, and the user information of electric vehicles is detected in real-time, thus ensuring the privacy security of electric vehicle users and the communication security of interactive data. Therefore, network attacks cannot steal the user's identity data and location privacy. Even if the attacker obtains the corresponding location information of the new energy vehicle, they cannot infer the user's basic information and the ciphertext's main content. The reason why the ciphertext cannot be obtained is that in the process of sharing the RSA public key and the RSA private key inside the charging pile between the user and the charging bank, the RSA encryption algorithm encrypts the charging pile exponentially and establishes PKS built-in trusted computing technology for physical isolation, authentication, trustworthiness, auditing and monitoring. It also makes it impossible for the charging rise of new energy vehicles to obtain the accurate information of anonymous users, which fundamentally realizes the data exchange and encryption between electric vehicle users and the charging pile. Only the RSA algorithm inside the charging bank can be used for decryption, thus ensuring the communication safety of data interaction between communication users. It provides privacy security in transmitting, storing and calculating private information such as identity, location, and preferences and ensures that authorized users can access the charging pile.

## 4. Topology Design of Charging Pile System

As shown in Figure 9, the system structure design and the main working principle of the automobile charging pile are composed of an AD/DC module, PFC controller and LLC harmonic corrector. In contrast, the control part comprises an input protection detection module, control unit, and communication module.

The system design structure diagram is shown in Figure 10. The core component of the AC charging pile control system is a single-chip microcomputer, which has a compelling signal processing ability and supports various peripheral devices. It is connected to a touch screen and indicator light through USART and I/O ports and is a vital function of human-computer interaction. The settlement module is mainly composed of two parts in hardware: an RFID card reader and a prepaid card. The reader interacts with the microcontroller through a serial peripheral interface (SPI). Users can use this interactive function to identify or pay charges. In the design of the power parameter measurement module, considering all aspects, it is mainly composed of an Ethernet connection to the serial port module and a built-in TCP/IP protocol stack [24]. They communicate with the microcontroller through an asynchronous serial interface (USART) to transmit data. The electrical protection module comprises a lightning arrester, AC contactor, emergency stop switch and circuit breaker. The microcontroller will also monitor the operational

status of the electrical protection module. During the monitoring process, the data such as the locking state, grounding continuity state and charging interface temperature of the charging gun are collected through an analog-to-digital conversion interface (ADC), and corresponding judgment is made according to the data. The I/O port controls the AC contactor, and the charging power supply is disconnected. To improve the anti-interference performance of the system, each functional module is designed independently in this paper, as shown in Figures 11 and 12.

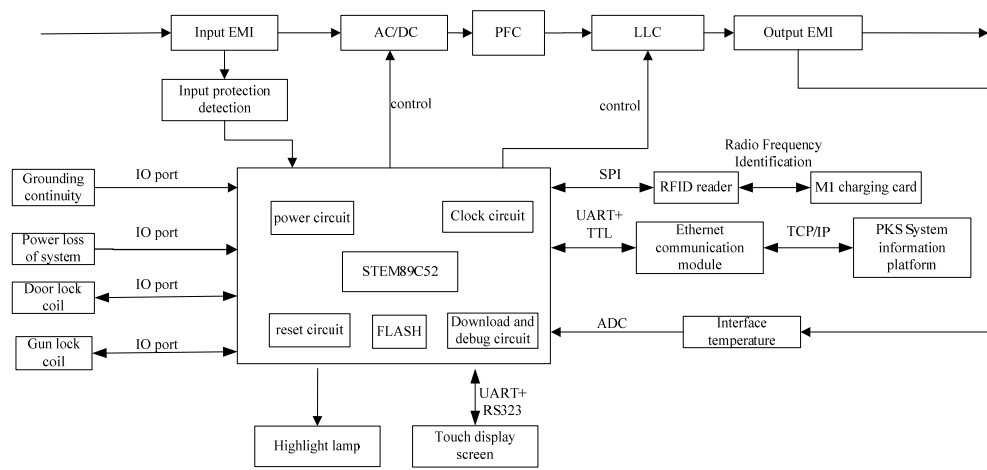

**Figure 9.** The structure design of the control system of the automobile charging pile.

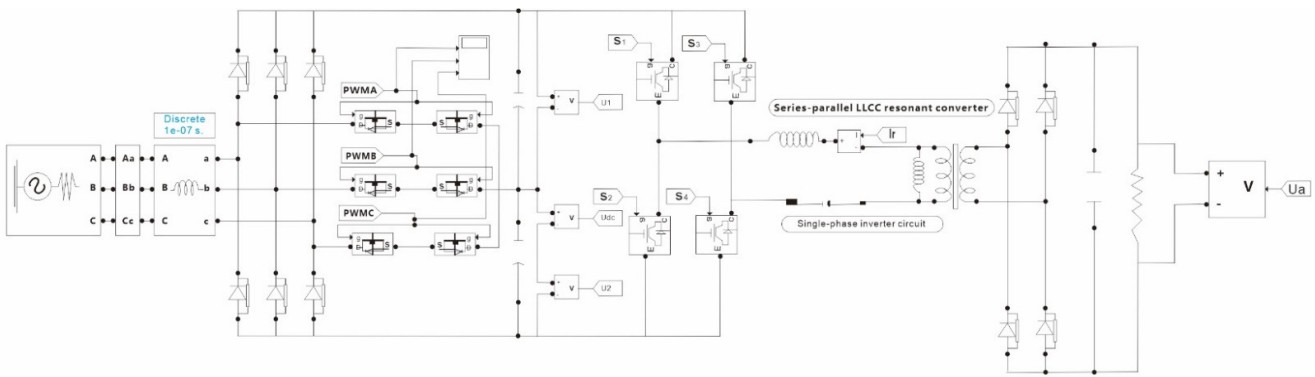

**Figure 10.** Design of power supply system for charging pile of new energy vehicle.

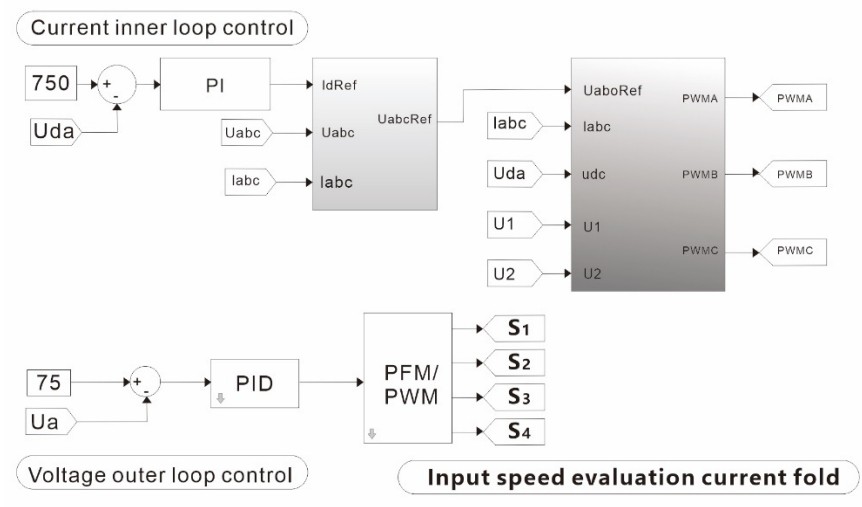

**Figure 11.** Input speed evaluation current fold.

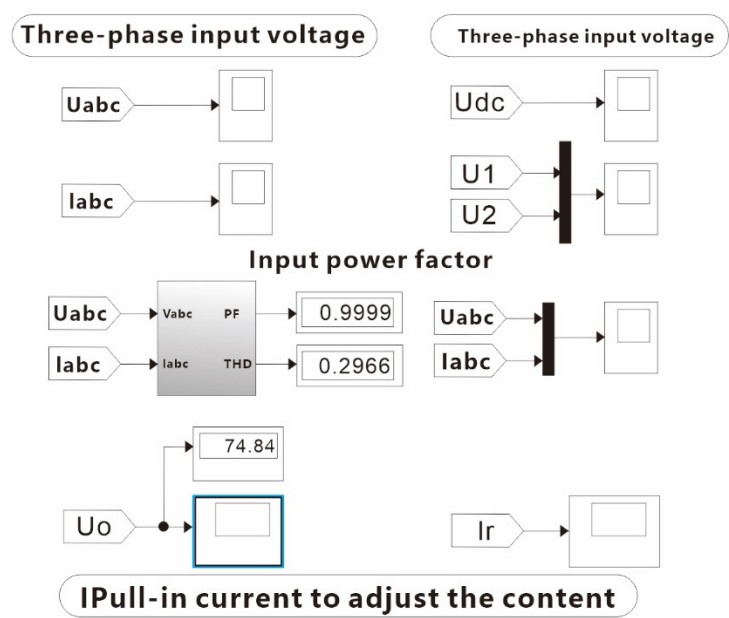

**Figure 12.** IPull-in current to adjust content.

According to the charging instruction sent by the client, it is encrypted by RSA and then decrypted by the PKS system operating system. After the encrypted charging instruction is decoded by the private key of the RSA key pair, the plaintext instruction of the charging instruction is distributed to the server system. Then, the charging pile system is controlled and converted into digital signals by compiling a microcomputer system to regulate the charging system. The AC380V standard three-phase voltage is input, and the bus voltage rises to DC750V after being handled by the three-phase bridge rectifier circuit and PFC. Then the LLC circuit topology structure generates AC square waves after the single-phase inverter IGBT is turned on and off [25], and the harmonic tank circuit filters the harmonic wave. After the original sinusoidal, square waves are converted into smooth sine waves by the high-frequency transformer, the output DC voltage is quickly stabilized to 75 V by the rectifier circuit and the low-pass filter circuit through the regulation of the PI + SVPWM control scheme, reaching the electrical level of domestic charging piles.

### 4.1. Design of Current Inner Loop PI Controller

The current loop system's structure diagram is shown in Figure 13. *TS*, $T_{ic}$ and $K_{ic}$ respectively represent the sampling period, the time constant of feedback and command current, and the current feedback coefficient; $K_{iI}$ is the integral coefficient, $K_{ip}$ is the proportional coefficient of the current loop controller, *IC* and *IC*\* are the output current and command current respectively. *US* and $K_{pwm}$, one is the power grid disturbance, and the other is the equivalent gain of *PWM*.

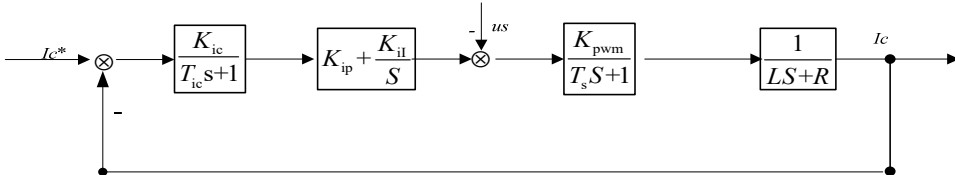

**Figure 13.** Structure diagram of current inner loop system.

Let the transfer function of the PI controller be Formula (5):

$$G_i(S) = K_{ip} + \frac{K_{iI}}{S} = \frac{K_{ip}(\tau s + 1)}{\tau_i s} \tag{5}$$

Among them. $\tau_i = \frac{K_{ip}}{K_{il}}$.

The voltage feedforward of the power grid can cancel out the power grid disturbance so that the power grid disturbance can be neglected in this case. The following expression can describe the open-loop transfer function of the current loop as Formula (6):

$$G_{Pic}(S) = \frac{K_{ic}Ki_pK_{pwm}(\tau_i s + 1)}{\tau_i s(s+1)(T_s s + 1)(LS + R)} \tag{6}$$

Let $T_{ic}$ = 0.5 $T_s$, $K_{ic}$ = 1, and the above formula can be simplified as Formula (7):

$$G_{Pic}(S) = \frac{K_{ip}K_{pwm}(\tau_i s + 1)/R}{\tau_i s(1.5T_s s + 1)\left(\frac{L}{R}S + 1\right)} \tag{7}$$

Make zero pole elimination, the open loop transfer function is Formula (8):

$$G_{Pic}(S) = \frac{K_{ip}K_{pwm}/L}{s(1.5T_s s + 1)} \tag{8}$$

The closed-loop transfer function of available current is Formula (9):

$$\phi_{ic}(s) = \frac{K_{ip}K_{pwm}/1.5LT_s}{s^2 + s/1.5T_s + K_{ip}K_{pwm}/1.5LT_s} = \frac{w_n^2}{s^2 + 2\xi w_n + w_n} \tag{9}$$

According to the second-order optimal design principle, the above formula can be converted into as Formula (10):

$$\xi = \frac{1}{3}\sqrt{\frac{1.5L}{T_s K_{ip}K_{pwm}}}, wn = \sqrt{\frac{KipKpwm}{T_s K_{ip}K_{pwm}}} \tag{10}$$

Make, can be obtained by substituting Formula (11); $\xi = 0.707$

$$K_{ip} = \frac{L}{3T_s K_{pwm}}, \; K_{il} = \frac{R}{3T_s K_{pwm}} \tag{11}$$

In this paper, the switching frequency and sampling period are 10 KHz and $T_s$ = 0.0001 s, and the inductance of the AC side and the equivalent resistance of the line are $L$ = 0.8 mH and $R$ = 0.05 $\Omega$, respectively. In the formula, the value of Kpwm is substituted, which takes 1, and the PI parameters of the current loop controller can be obtained by calculation: $K_p$ = 2.67, $K_i$ = 166.67. As Formula (12)

$$D_i(s) = 2.67 + \frac{166.7}{S} \tag{12}$$

Substituting (12) into Formulas (6) and (7) to obtain.
Open loop transfer function as Formula (13):

$$G_{vc}(s) = \frac{K_{vp}(\tau_v s + 1)}{C\tau_v s(s+1)} \tag{13}$$

Closed loop transfer function as Formula (14):

$$\phi_{ic}(s) = \frac{1}{4.5T_s^2 s^2 + 3T_s s + 1} \tag{14}$$

As shown in Figures 14 and 15, the open-loop transfer function of the existing outer loop shows the phase angle margin of the present inner loop.

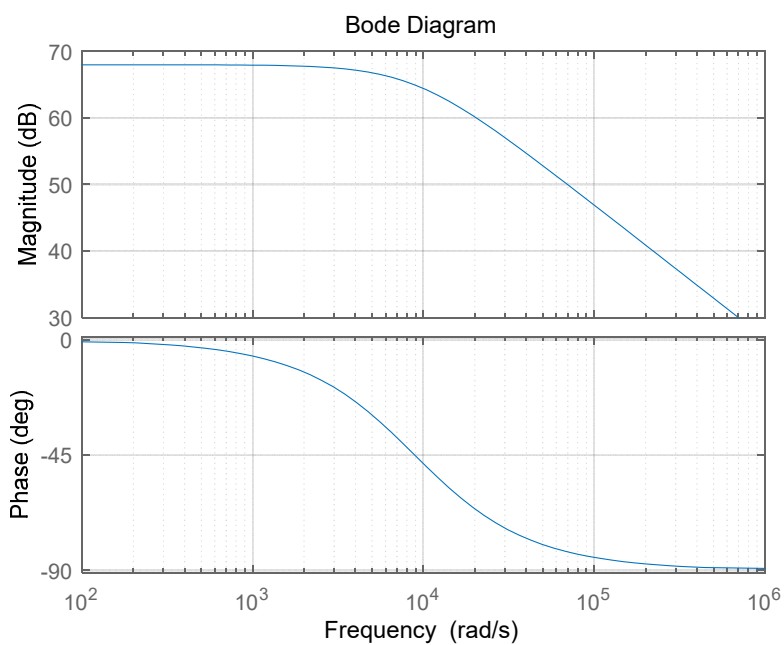

**Figure 14.** Bird diagram of the open loop transfer function of the current outer loop.

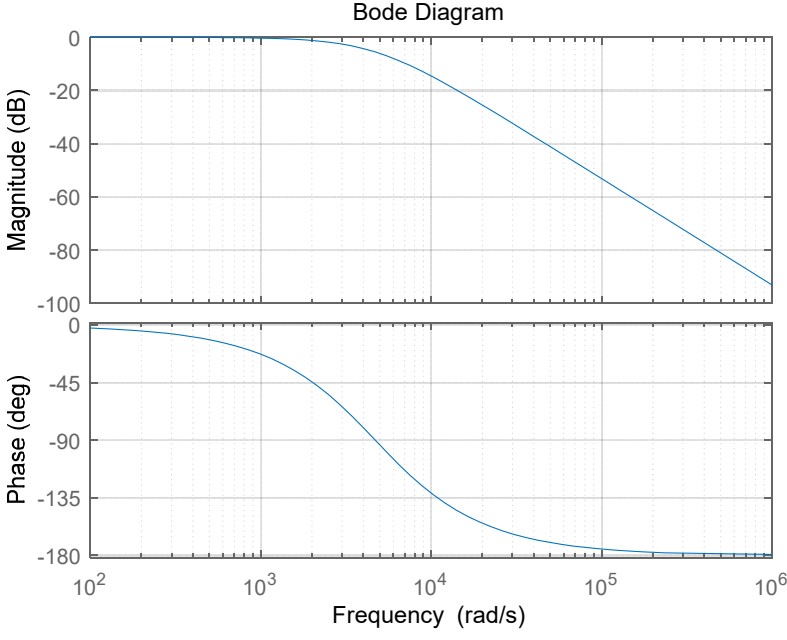

**Figure 15.** Bode diagram of a closed-loop transfer function of the current outer loop.

### 4.2. Design of PI Controller for Voltage Outer Loop

The voltage outer loop can control the DC voltage. Still, since the capacitor voltage UC will change continuously during charging, it is not static, so when generating the compensation current, part of the active power should be used to maintain the DC voltage stability. The reference DC voltage is input through *udc\**, and the DC voltage *udc* is output through the transformation process. The PI controller that outputs the DC voltage is shown in Figure 16.

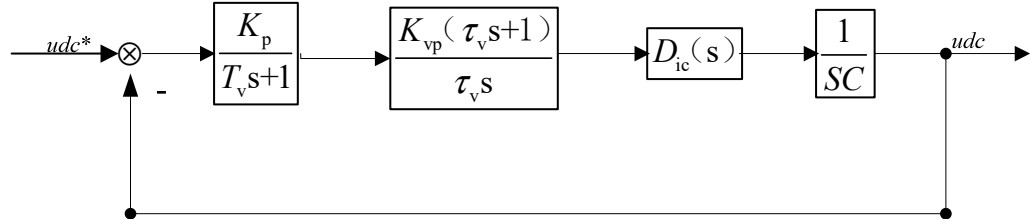

**Figure 16.** Structure Diagram of Voltage Outer Loop Control System.

The feedback gain $k_V$ is 1, the feedback delay time constant $T_V$ is 1, and the output DC side capacitance c is 1. Combine the first-order inertia link of the current loop transfer function with the feedback link to form a new inertia link. Therefore, the open-loop transfer function of the voltage outer loop transfer function is Formula (15):

$$G_{vc}(s) = \frac{K_{vp}(\tau_v s + 1)}{C\tau_v s(\tau_h s + 1)} \tag{15}$$

The closed-loop transfer function of the voltage outer loop transfer function is Formula (16):

$$\phi_{vc}(s) = \frac{K_P K_{vp}(\tau_v s + 1)D_{ic}(s)}{SC(\tau_v s + 1)\tau_v s + K_p K_{vp}(\tau_v s + 1)D_{ic}(s)} \tag{16}$$

According to the typical second-order system design as Formula (17):

$$\frac{K_{vp}}{C\tau_v} = \frac{h_v + 1}{2h_v{}^2 T_h{}^2} \tag{17}$$

Take $h_v = 5$ and substitute it into the above formula to obtain as Formulas (18) and (19):

$$K_{vp} = \frac{3C}{5T_h} \tag{18}$$

$$K_{vi} = \frac{3C}{25T_h{}^2} \tag{19}$$

The sampling period and voltage loop delay time constant are $10^{-4}$ s and $10^{-4}$ s, respectively. $C = 0.0005$ F, and substituting it into (18) can calculate: $K_i = 0.1$, $K_P = 0.0002$.

The transfer function of voltage outer loop PI controller as Formula (20);

$$D_{ev}(s) = 0.0002 + \frac{0.1}{s} \tag{20}$$

### 4.3. SVPWM Mode of Active Power Filter System

Active power filter PWM drive strategy sinusoidal pulse width modulation (SPWM) and voltage space vector pulse width modulation (SVPWM) [26]. Compared with the former driving strategy, the latter has the advantages of low harmonics, simple digital control mode and high voltage utilization rate. Therefore, this driving strategy will be used in most cases. The following system analyzes the single-phase SVPWM.

Each switch state of the single-phase inverter circuit has an output, and the specific structure is as Figure 17:

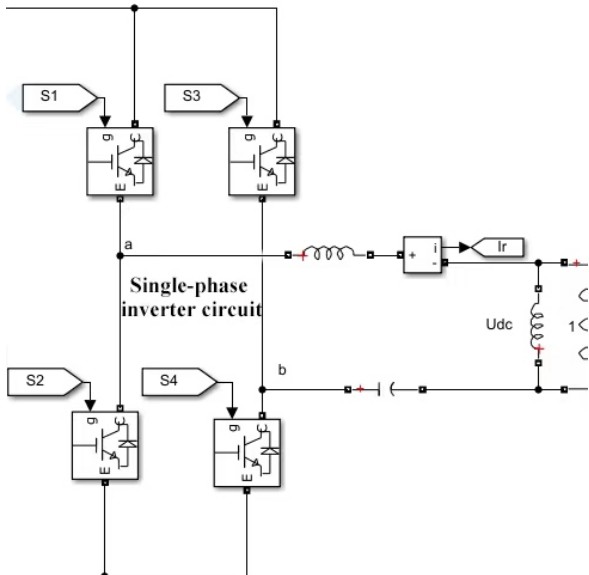

**Figure 17.** Single-phase inverter circuit.

The four working states of the full-bridge inverter circuit are shown in Table 2:

**Table 2.** Four active forms of the full-bridge inverter circuit.

| Sa | Idot | Uab | Correspondence |
|---|---|---|---|
| 0 | 0 | 0 | V0 |
| 0 | 1 | $-Udc$ | V1 |
| 1 | 0 | $-Udc$ | V2 |
| 1 | 1 | 0 | V3 |

The voltage space vector corresponding to the four switching states of the single-phase inverter has a negative vector and a positive vector, so the modulated UreF voltage only moves on the positive and negative half shafts.

If the sampling period is $T_s$, Reef is in the positive half-axis. Then there are Formula (21):

$$U_{\text{refTs}} = U_1 T_1 + U_0 T_0 \tag{21}$$

As Formulas (22) and (13) can:

$$T_1 = \frac{U_{\text{ref}} T_s}{V_1} = \frac{U_{\text{ref}} T_s}{V_{\text{dc}}} \tag{22}$$

$$T_0 = T_s - T_1 \tag{23}$$

If Reef is greater than or equal to zero, then $T_1$ and $T_0$ times of $U_{\text{ref}}$ (1 1,0) are similar.

If Reef happens to be in the opposing half-axis, and if the sampling period is Ts, then there are Formula (24):

$$U_{\text{refTs}} = V_2 T_2 + V_0 T_0 \tag{24}$$

As Formulas (25) and (26) can:

$$T_2 = \frac{U_{\text{ref}} T_S}{U_2} = \frac{|U_{\text{ref}}| T_S}{U_{\text{dc}}} \tag{25}$$

$$T_0 = T_s - T_2 \tag{26}$$

If Reef is less than zero, then $T_2$ and $T_0$ of $U_{\text{ref}}$ (1 1,0) are equal.

Therefore, the duration of each switching cycle of the two switching states can be calculated by the following two Formula (27):

Effective time:

$$T_x = \frac{|U_{\text{ref}}|T_S}{U_{\text{dc}}}$$ (27)

Zero vector time as Formula (28):

$$T_0 = T_s - T_2$$ (28)

### 4.4. Comparison of Circuit Topology System

*Option 1*

Power conversion is a vital link to determine the charging link of the charging pile, and the simplest power conversion topology is the first-order topology. Only one-stage topology is used to realize the functions of PFC and output voltage regulation. As shown in Figure 18, they have a common feature: the AC side needs two switch tubes in anti-parallel to form a bidirectional switch. The disadvantage of this scheme is that the input current is in phase with the input voltage, and the input power changes to twice the energy frequency, so only the output power wave is more significant. All AC side switches need to flow with a high-frequency current [27]. Under complex working conditions, the robustness of this scheme is poor, and there is no mature application product at present.

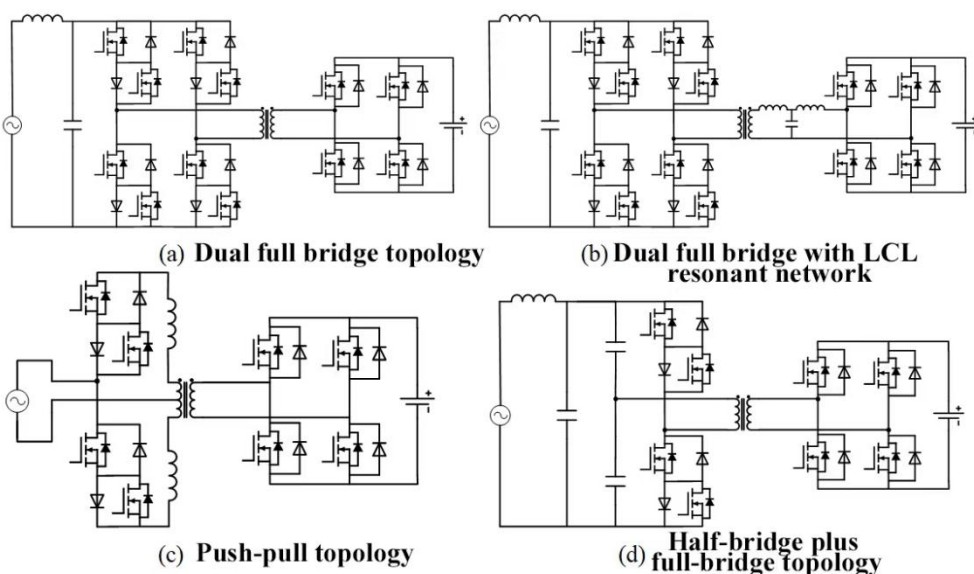

**Figure 18.** Output-side equivalent circuit of first-level topology and first-level topology scheme.

*Option 2*

The two-stage scheme is the most mature scheme at present. Usually, the front stage is PFC topology with bidirectional energy flow capability, and the backstage is isolated DC/DC. Power factor correction is mainly the responsibility of the previous step. The extensive bus capacitance completes two-stage decoupling in the middle, and the steady-state regulation is conducted later. Its advantages are that the two-stage circuit control is decoupled, the dynamic performance is good, the output power frequency ripple is small, the two-stage efficiency can be optimized to a higher level respectively, and better robustness can be obtained in engineering applications; The defect requires large bus capacitance, which is the difficulty to optimize the volume further. In the two-stage scheme, because of the large electrolytic capacitor of the bus, it can buffer twice the power frequency. Similar to the previous analysis method, the preceding PFC can be equivalent to a current source with twice the power frequency ripple. Figure 19 shows the equivalent circuit.

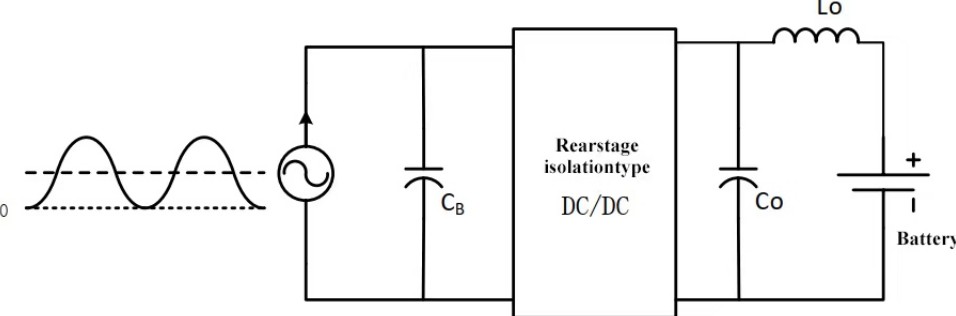

**Figure 19.** Equivalent circuit of the output side of two-stage topology scheme.

The two-stage scheme mainly relies on the large capacity DC bus capacitor to reduce the output ripple at the double power frequency, and the key to its optimization lies in increasing the bandwidth of the post-stage system and the control gain at the essential characteristics of the output double-power frequency ripple and does not need to pay the extra hardware cost.

In sum, it is generally believed that the two-stage scheme is more advantageous: the front stage PFC+ and the backstage DC/DC.

## 5. Simulation Experiment

In this paper, two simulation experiments of the charging system are carried out. Firstly, the security of the charging system based on the PKS System is analyzed. By comparing with Intel + Linux System, it is concluded that the protection area of the PKS System is better than that of the Intel + Linux System under multi-process and multi-means test conditions. The RSA encryption algorithm ensures the security and privacy protection of the software communication at the upper level of the system and effectively prevents attacks and intrusions. At the status of the charging pile hardware system, the simulation verification of space vector modulation for the topological charging pile system proves that the proposed topology improves the charging efficiency and energy consumption loss.

### 5.1. Security Verification Based on the PKS System

PKS is an integrated design and system from the bottom chip to the upper application. At the same time, it standardizes the security interface standard so that all security control points can be connected efficiently and cooperate tacitly and controlled by a unified control center. The protection effect is better. After the network attack test, PKS's protection ability is better than that of similar foreign traditional protection products of Intel + Linux under vulnerability triggering, load execution, downloading the main body, main body execution, local power raising and back door curing. The comparison results are shown in the following Table 3.

The PKS system is based on the eBPF [28] technical architecture, which can run sandbox programs in the operating system kernel so that the kernel functions can be extended safely and effectively, and ePBF is applied to the network processing module. The combination of programmability and efficiency makes eBPF a natural fit for all packet processing requirements of networking solutions. The programmability of eBPF enables adding additional protocol parsers and easily programming any forwarding logic to meet changing requirements without ever leaving the packet processing context of the Linux kernel. Therefore, using and processing encrypted messages in the network card ensures that the data security and charging system core is also based on ePBF technology. Building on the foundation of seeing and understanding all system calls and combining that with a packet and socket-level view of all networking operations allows for revolutionary new approaches to securing systems.

**Table 3.** Comparison of PKS and traditional protection capability.

| Means | Intel + Linux | | | | | | PKS | | | | | | R SA |
|---|---|---|---|---|---|---|---|---|---|---|---|---|---|
| **Process** | Trigger vulnerability | Load Running | Download Main Body | Subject Execution | Local Empower-ment | Backdoor Curing | Trigger Vulnera-bility | Load Running | Download Main Body | Subject Execution | Local Empower-ment | Backdoor Curing | R SA |
| Data area code execution | (red) | (green) | | | | | (red) | (green) | | | | | (yellow) |
| Stack overflow-turn on stack protection | (red) | (green) Protective Area | | | | | (red) | (green) Protective Area | | | | | (yellow) |
| Integer overflow-turnon stack protection | (red) | (green) | | | | | (red) | (green) | | | | | (yellow) |
| Kernel UAF | (red) | (green) | | | | | (red) | (green) | | | | | (yellow) |
| Double Free | (red) | | | | | | | | (yellow) | (yellow) | (green) | | (yellow) |
| Stack overflow-wireless protection | (red) | | | | | | | | (yellow) | (yellow) | (green) | | (yellow) |
| Overflow-wireless protection | (red) | | | | | | | | (yellow) | (yellow) | (green) | | (yellow) |
| Application Layer UAF | (red) | | | | | | | | (yellow) | (yellow) | (green) | | (yellow) |
| Memory Attack | (red) | | | | | | | | (yellow) Enhanced Area | | | | (yellow) |

In the attack scenario, ePBF controls and monitors system processes and module units, as shown in Figure 20. eBPF runs on the network card module. It can filter unsafe data packets and pass them to the kernel for execution. The filtering process is also a monitoring process. This process can ensure that the encrypted data cannot be physically attacked during transmission; if the data is contaminated, the monitoring system will activate the early warning mechanism to load the alarm code to execute the alarm event, which improves the reliability of the data in the encrypted communication process.

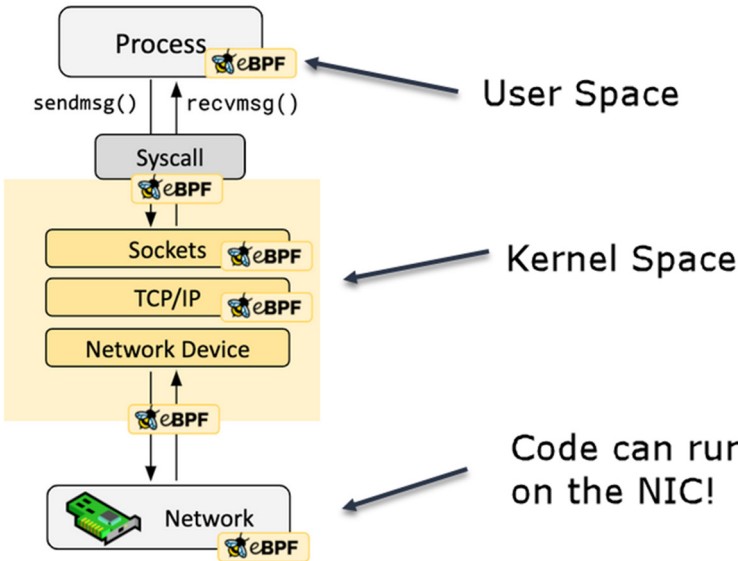

**Figure 20.** The running process of ePBF in the network card.

Based on the analysis of the security performance of user information flow, users' private data and car-machine system data often leak in the interaction between users and charging piles. We will involve mechanisms from the interaction between the integrated service layer and hardware. Here, we use an encryption algorithm to encrypt plaintext data further. Here, we only encrypt user information and transaction data utilizing mechanism encryption. A suitable mechanism is only the beginning of encryption, and the key depends on the system support of the charging pile's data platform system. After the network attack test, the performance indicators of vulnerability triggering, load execution, downloading subject, subject execution, local power raising and backdoor curing are tested. After the network attack test based on the RSA encryption algorithm, the performance indicators of vulnerability triggering, load execution, downloading subject, subject execution, local power raising and backdoor curing are tested. However, after the network attack test based on the RSA encryption algorithm, the protection area is more significant than without the RSA encryption algorithm.

### 5.2. Simulation Verification of Space Vector Modulation

In this paper, MATLAB SIMULINK software is used to verify the proposed system, and the trigger signal of MOSFET is simulated. By inputting the Iabc three-phase current and UabcRef three-phase reference voltage. SVPWM modulation can reduce switching power consumption to a certain extent. See the following Figure 21 for details of the SVPWM modulation simulation module:

As shown in Figure 22, The PI controller controls it, then headed by Uabc and Iabc through the phase-locked loop to refer to Uabc, and then modulated by SVPWM with Iabc, Udc, U1 and U2 to output modulation signals PWMA, PWMB and PWMC to turn on and off MOSFET tubes. The SVPWM modulation module simulation can transform the trigger signal from the original sinusoidal signal with noise into a flat trapezoidal wave, reduce the power consumption of MOSFET tubes, and more accurately control the cycle of each

trigger signal of MOSFET tubes to avoid the stagger of trigger time. As shown in Figure 23. The color of each line represents the waveform display of the PWM control voltage with the phase shift of the three-phase voltage differing by 120 degrees. The PWM waveform is shown in Figure 24, The color of each line represents a 380 V AC voltage with a phase shift of 120 degrees between the input three phases.

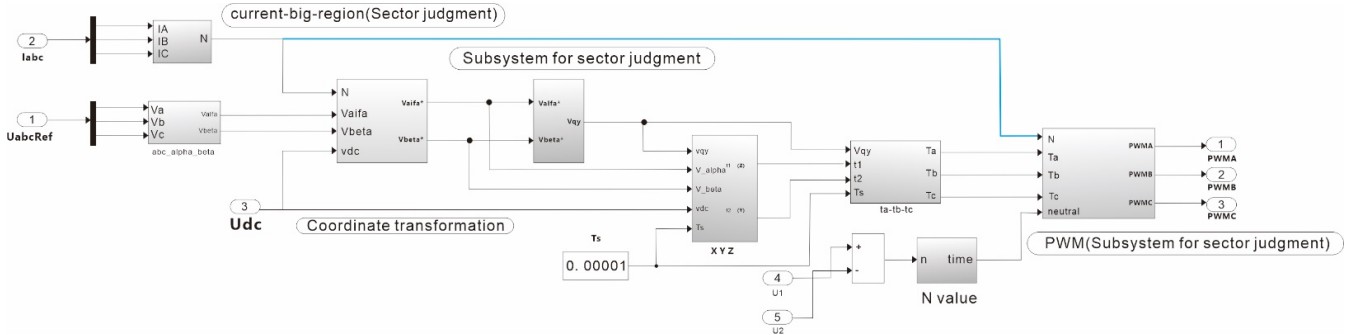

**Figure 21.** Simulation diagram of SVPWM modulation module.

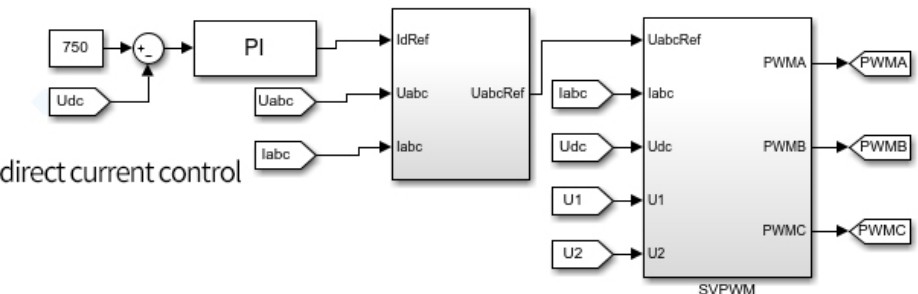

**Figure 22.** PWM model of PI control and SVPWM control.

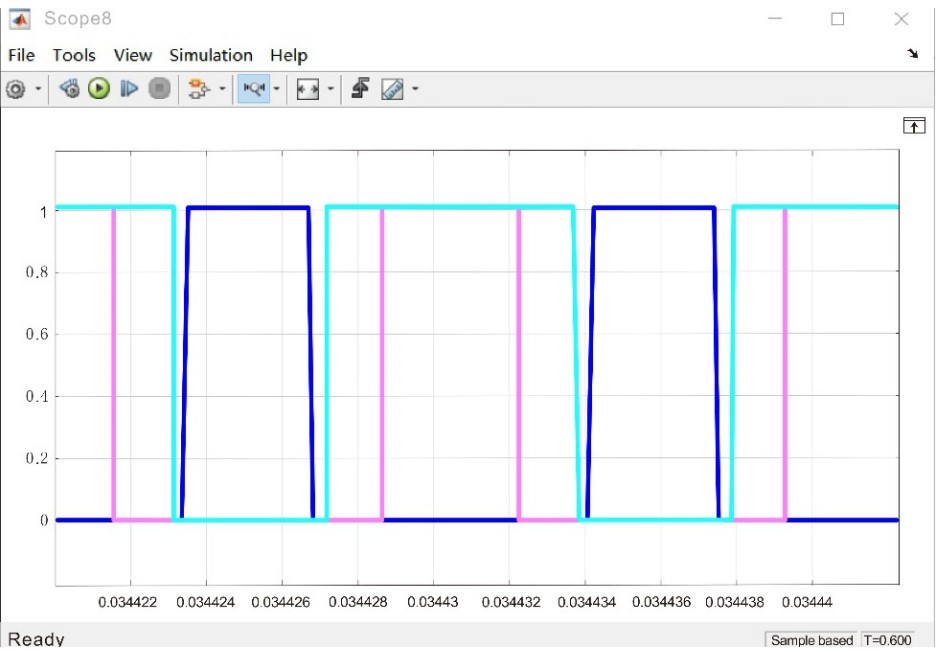

**Figure 23.** The waveform of the modulated PWM control signal.

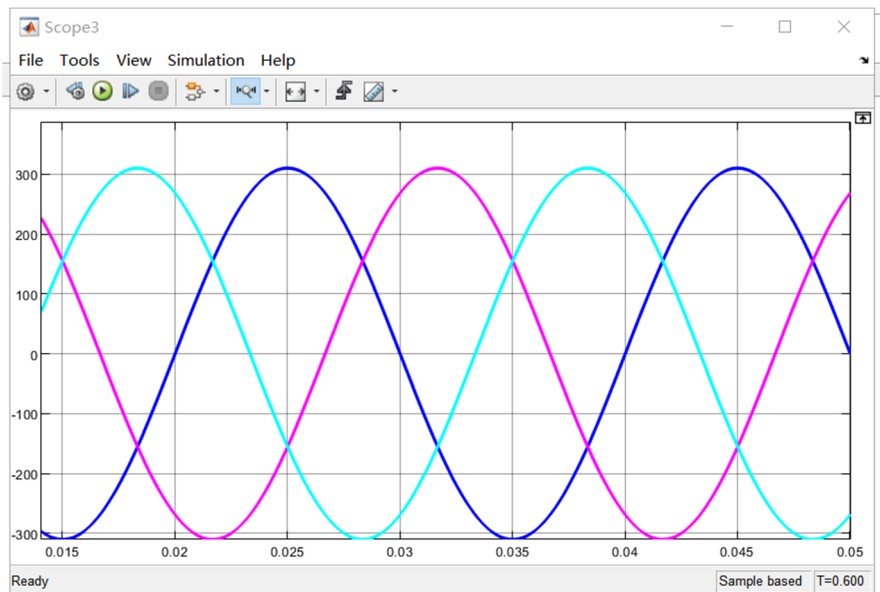

**Figure 24.** Simulation display diagram of three-phase input voltage.

The distortion rate of the input current is 1.81%, which shows that the SVPWM technology + PI control strategy scheme is feasible and can effectively restrain the current distortion and generate higher harmonics, thus reducing the waste of electric energy and switching loss, thus improving the output efficiency and the power factor is close to 1. Through the verification of the above simulation results that the causes of component loss are spurs and distorted sinusoidal waveforms.

This is the simulation diagram of the three-phase input voltage. The input voltage is AC 330 V and 50 HZ, and the phase difference between the three phases is 120 degrees. Moreover, the three-phase AC is the same as the phase of the voltage.

After the rectification of the three-phase fully-controlled rectifier circuit, there is also a PFC power corrector in the middle to adjust the voltage and current of the three-phase input to the same phase and adjusts the original distorted current to a smooth sine wave, which reduces the switching loss and stabilizes the bus voltage to 750 V.

The AC 380 V three-phase voltage input is adjusted by a three-phase bridge rectifier circuit and PFC, and then the bus voltage rises to DC750V. Then, in LLC topology, the AC square wave is generated by the turn-on and turn-off of a single-phase inverter IGBT [29], and then the harmonic wave is filtered by a harmonic tank circuit. After the original sinusoidal, square wave is converted into a smooth sine wave by a high-frequency transformer, the output DC voltage is quickly stabilized to 75 V through the adjustment of the rectifier circuit and low-pass filter circuit by PI + SVPWM [30] control scheme.

As Figures 24 and 25 are the simulation diagrams of the three-phase input voltage and current, respectively, where each color in the simulation diagram of the three input currents in Figure 25 represents the current on a different phase. Corresponding to the colors in Figure 24, respectively.

The simulation model diagram (Figure 9) is set up in this paper, and the software MATLAB SIMULINK is used to build this diagram. In Figure 26, the input 380 V AC with a frequency of 50 HZ is converted into the 750 V voltage of the DC side Udc by the three-phase rectifier of the front stage circuit. After passing through the full-bridge LLC resonant circuit converter of the rear stage circuit (the resonant frequency is 150 kHz, the PWM is controlled by frequency conversion, and the PWM is controlled by variable duty ratio), the output current and voltage are closed-loop controlled by the current inner loop PI and the voltage outer loop PI, so that the output voltage is 24 V–75 V, The rated power is 12 KW, the working efficiency is 95%, the input power factor is close to 1, and as Figure 27 the current harmonic content is 1.8%. As Figure 26 the bus voltage of the rectified capacitor

are shown. As Figure 28 the simulation waveform of the output voltage simulation diagram is close to the theoretical calculation value, proving the scheme's feasibility and practicality.

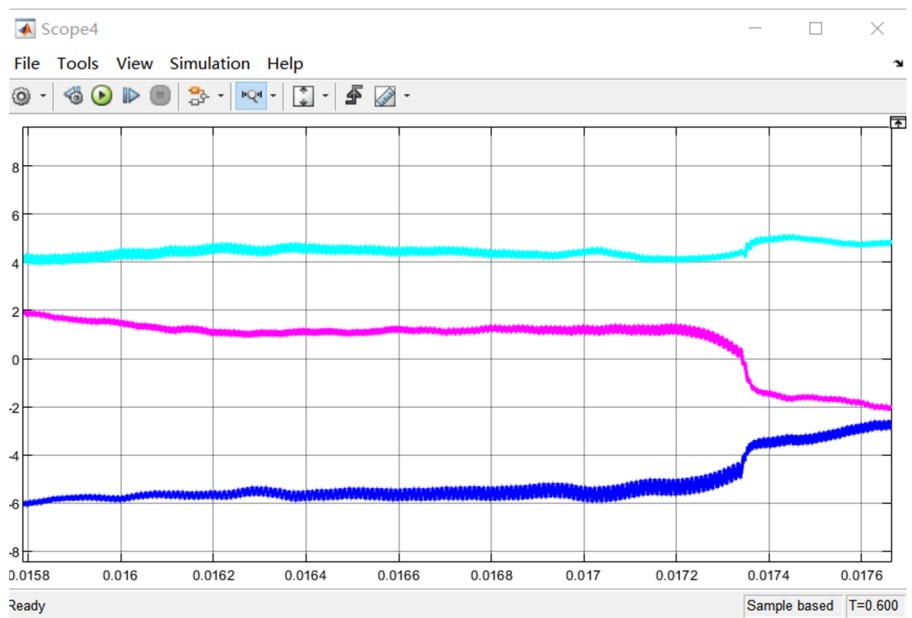

**Figure 25.** Simulation display diagram of the three-phase input current.

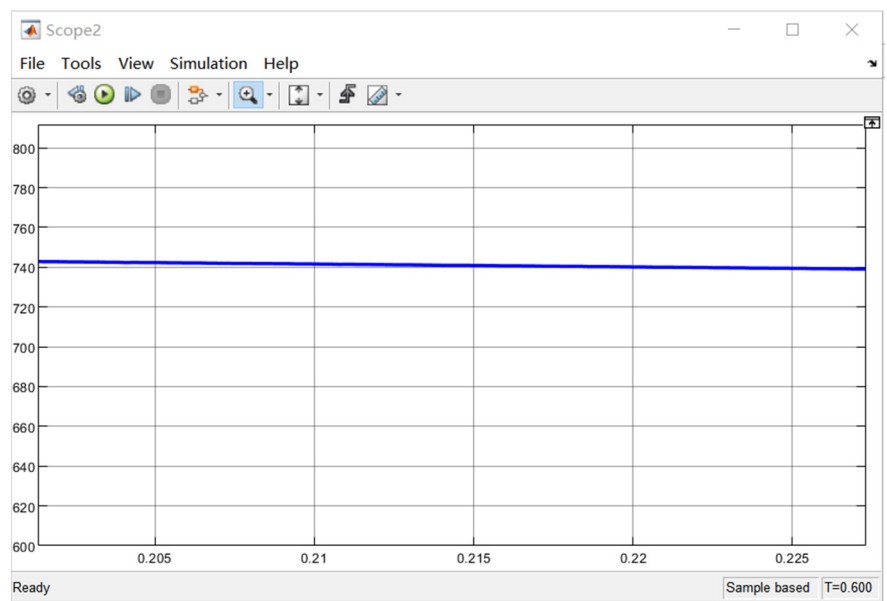

**Figure 26.** Simulation display diagram of bus voltage of capacitor after re ctification.

*5.3. Experimental Results*

In this paper, the practical test of the secondary circuit design based on the topology structure is done. By setting the workflow in the circuit to AC380V, the power frequency 50 HZ is input after EMI is eliminated to eliminate electromagnetic interference. Then the three-phase fully-controlled rectifier circuit is used for the voltage value after rectification. It will increase the influence of the nonlinear characteristics of components.

To avoid harmonics entering the grid and unnecessary switching and power loss, a PFC (power factor corrector) is added behind the three-phase rectifier bridge to make the power factor close to 1. The efficiency of electric energy use is improved. The subsequent stage adopts BOOST LLC continuous type, and the inverter voltage obtained by PFC is

750 V. The opening and closing of the IGBT in the inverter circuit is an AC square wave that goes through the harmonic circuit and is converted by a high-frequency transformer. It is a smooth sine wave and then eliminates harmonic pollution through EMI and enters the grid to charge the battery. The PCB experimental circuit board is shown in the figure, Figure 29 is the PFC engineering prototype, and Figure 30 shows the LLC engineering prototype. The designed secondary topology circuit can be highly integrated into a PCB circuit board in the production process.

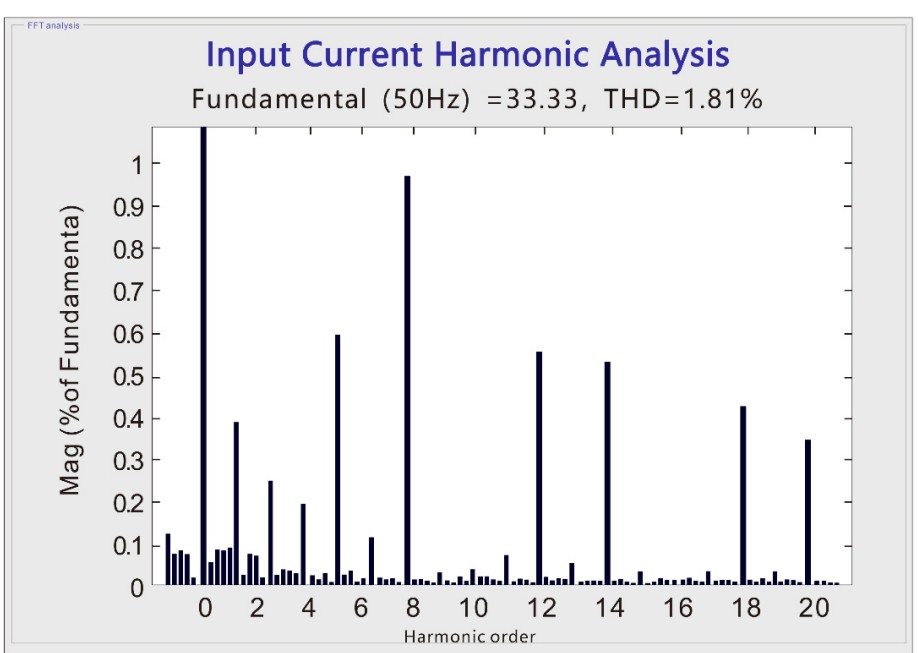

**Figure 27.** Harmonic analysis of input current.

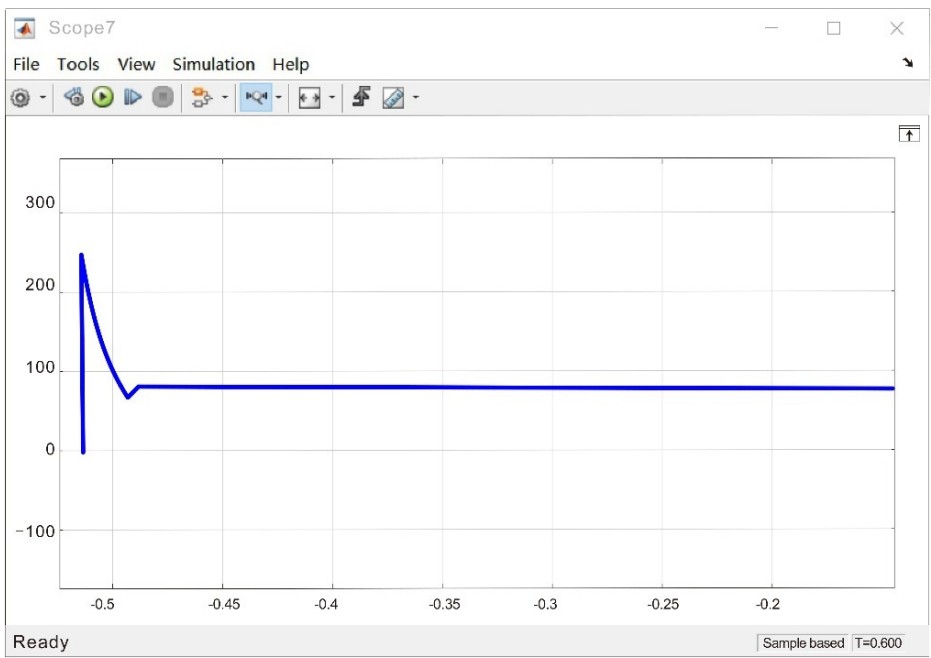

**Figure 28.** Simulation diagram of output voltage.

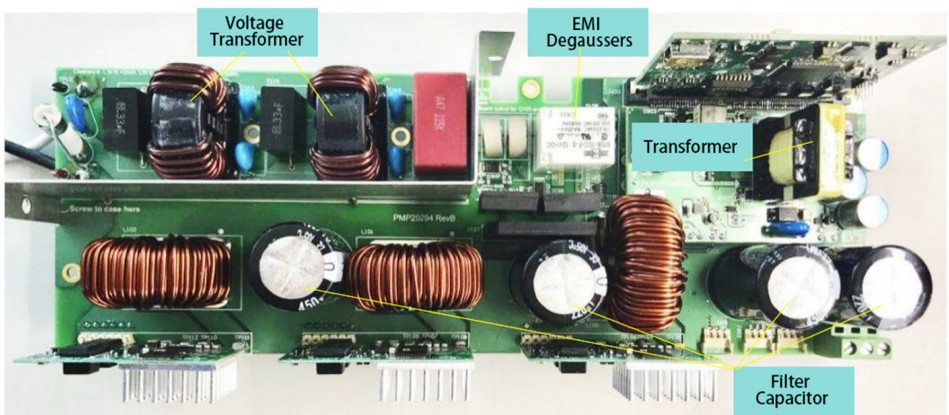

**Figure 29.** Level 1 PFC engineering circuit board.

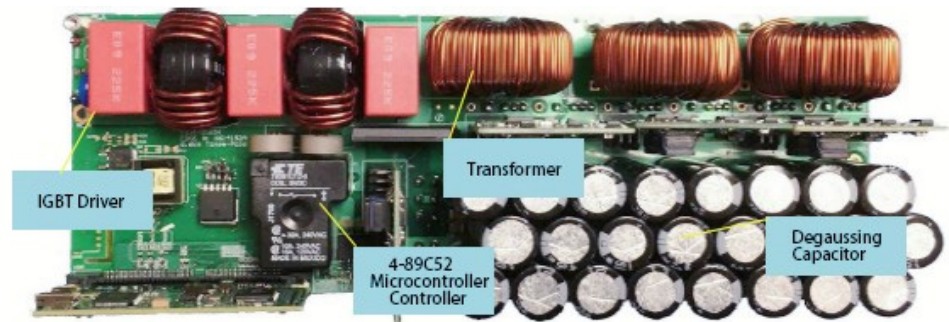

**Figure 30.** Level 2 LLC rectifier engineering circuit board.

As shown in Figure 29, the PFC circuit board plays the role of alternating current to direct current, the filter capacitor element on the board is used for noise reduction and filtering, and the copper coil is used for voltage transformation.

As shown in Figure 30, the LLC rectifier circuit plays the role of DC to AC and AC to DC. Adjusting the current, the rectified voltage tends to the theoretical output value.

In the experimental results, the bus voltage rose to AC750V after adjustment by PFC, and the output voltage waveform is stable, as shown in Figure 31. However, some factors affect the accuracy of the experiment during the experiment, such as the power drop of some devices caused by artificial solder joints, the quality of components, the problem of winding turns ratio, and the noise problem when the filter capacitor stores the voltage—all of which affect the stability of the output voltage. Then, the output 75 V DC voltage was adjusted through the high-frequency transformer, and five effective measurement results were taken after multiple measurements. As shown in Table 4, the experimental voltage is about 75 V, and the power is about 12 KW.

**Table 4.** Output voltage current and power measurement results.

| Times | Voltage (V) | Current (A) | Power (KW) |
|:-----:|:-----------:|:-----------:|:----------:|
| 1 | 74.1 | 146.42 | 10.85 |
| 2 | 75.2 | 150.93 | 11.35 |
| 3 | 73.8 | 159.21 | 11.75 |
| 4 | 75 | 159.33 | 11.95 |
| 5 | 75.4 | 159 | 11.989 |

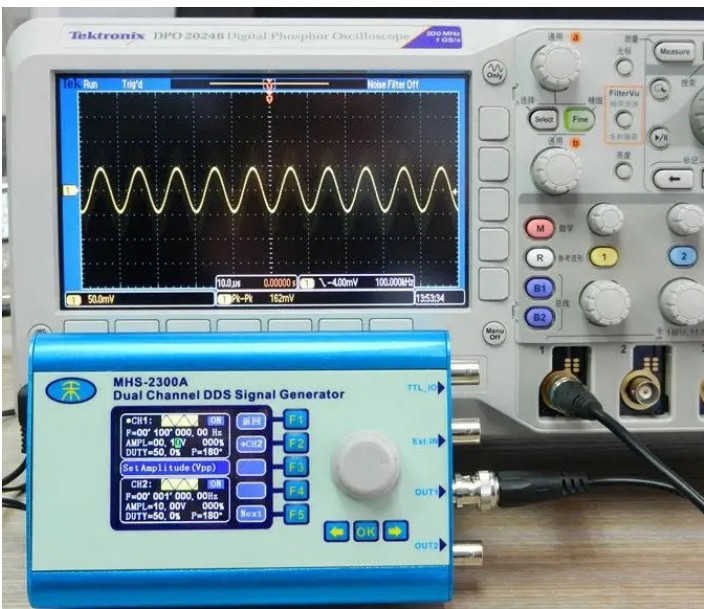

**Figure 31.** Output bus voltage waveform.

## 6. Discussion

In the rapidly developing new energy vehicle charging environment [13,15], the problem of mileage anxiety and the problem of data security in the big data environment is worrisome. This paper improves the charging efficiency by improving the topology structure and introduces the RSA encryption algorithm to ensure communication security. Experiments verify the proposed method. In the fifth part, the judgment method and comparison of the trustworthiness and security of the data encryption communication system and the topology circuit are described, respectively. Simulation and engineering experiments show the feasibility and importance of the proposed architecture. The proposed architecture can provide a reference in the future field of new energy vehicle design and application control systems. The data communication encryption method in the proposed architecture has not solved the brute force-cracking of the encrypted data in the client IC card. Due to the limitations of the experiment, some factors affected its accuracy, such as the power drop of some devices caused by artificial solder joints, the quality of components, the problem of winding turns ratio, and the noise problem when the filter capacitor is stored, which will affect the conversion efficiency. Future work in this regard should include:

1. Future research on the scalability of the proposed charging system;
2. Further optimizing of the cells in the topology, as it reduces the number of components.
3. Integration of the charging system of the architecture and actual environmental stress tests.
4. An improvement of the quality of experimental components and re-layout of the board.

## 7. Contributions

This paper proposes the topology structure and secure encrypted transmission of the charging system and improves the overall architecture of the charging pile system.

1. The topological structure is proposed to improve the charging pile system. The conversion efficiency is improved by the unit components in the topological structure and the reduction of circuit noise.
2. An encryption algorithm is introduced in the communication process of the charging system to encrypt the circuit control information and data information in the communication link to ensure the safety of the charging system.
3. The experimental and simulation experimental data are obtained through the simulation experiment and the simulation output voltage of the PCB engineering circuit board in the laboratory environment.

## 8. Conclusions

This paper proposes a new energy vehicle charging pile system and communication network information security based on the PKS System. The simulation and debugging of the power charging unit of the proposed system make PWM reduce the switching loss to a certain extent. We used a double closed-loop current and voltage plus PI algorithm to control DC voltage and harmonic current to determine whether the power debugging scheme improves the energy loss problem. It enhances energy conversion efficiency, which can meet the requirements of basic electrical technical tasks. In terms of communication security and privacy protection, the encryption algorithm based on RSA is proposed, ensuring information interaction security in the charging network system and reducing the risk of privacy information leakage. Comparing the network security system based on PKS with Intel + Linux shows that the PKS System based on the RSA encryption algorithm offers the best protection. In an ideal environment, the simulation and testing of new energy vehicle charging may lead to some data deviation and a limited choice of decision-making strategy, which may lead to some power loss and influence data transmission stability in the process of charging system control and data transmission. As the PKS System is still in the experimental stage, the security based on the PKS System and the efficiency of the whole charging system will be improved in the future. To improve stability, the system of the software transmission level and hardware power control levels should be enhanced.

**Author Contributions:** J.Z.: conceptualization, writing—original draft preparation, writing—review and editing; J.L.: software; Y.X.: conceptualization, formal analysis, investigation; Y.Z., W.L. and S.Z.: validation; L.Z.: supervision; S.Z.: project administration, funding acquisition. All authors have read and agreed to the published version of the manuscript.

**Funding:** Finance Science and Technology Project of Hainan Province (Project Number: ZDKJ 2020009).

**Data Availability Statement:** Not applicable.

**Conflicts of Interest:** The authors declare no conflict of interest.

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
