# Peer review of "The Design of a Safe Charging System Based on PKS Architecture"

_electronics, doi:10.3390/electronics11203378_

Round 1

Reviewer 1 Report

The paper proposes an efficient charging and privacy protection system based on the PKS system. By improving the circuit topology, the loss of the charging pile is reduced, the conversion efficiency is improved, and the technical requirements for low energy consumption are met. The private data in the shared information uses the RSA encryption algorithm to ensure private data's leakage and enhance system communication's security. This paper aims to improve the charging efficiency of charging piles and the security of private information in network communication. Both simulation and experiential results are presented. So generally the paper presents well-prepared research. It's in the scope of the Electronics journal. I belief this topic is worth to be publised.

But I have some recommendations to improve it:

-> in abstract main results from both simulation and experimental studies should be presented in quantitative approach. Please revise.

-> Figure 1 is a table not figure. So please change to table.

-> the literature review for this paper is out of date. Please include at least 3 positions from 2022 year.

-> Figure 10 "Design of Power Supply System for Charging Pile of New Energy Vehicle" should be one-figure type in accordance to the MDPI temple. In the present form it's to small. The same issue fith fig 19 " Simulation Diagram of SVPWM Modulation Module"

-> please add a separate section (no 6) with a discussion of the results in the context of secondary literature.

Reviewer 2 Report

In this paper, an efficient charging and privacy protection system based on the PKS system is proposed. By improving the circuit topology, the loss of the charging pile is reduced, the conversion efficiency is improved, and the technical requirements for low energy consumption are met. The private data in the shared information uses the RSA encryption to ensure private data's leakage and enhance system communication's security. The work aims to improve the charging efficiency of charging piles and the security of private information in network communication. Simulation experiments are carried out through the proposed system scheme combining hardware topology and software encryption. The experimental results are given by the authors to show that the proposed system is feasible and safe.

There are some issues which need to be handled by the authors.

1.      On page 9, This formula can also- is incomplete.

2.      Explain Figure 7 components wise. For example CA is not explained anywhere in the text.

3.      Make Figure 1 as Table 1.

4.      On 2nd page (in last paragraph), replace Chapter by Section.

5.      Follow the uniformity throughout the paper. For example- in text, Figure should be Figure only not figure.

6.      In Conclusion section, the first line contains System two times.

Reviewer 3 Report

The authors of the paper studied methods to design safe charging system for new energy vehicles. The topic is interesting and the paper is well-structured. Some issues should be corrected and explained in a revision.

- The abbreviations in the keywords should use full spellings. 

- The contribution of this work should be itemized and highlighted.

- In Fig. 3, what is P+K?

- In addition to simulations, have you implement the designed circuits and carried out real-world experiment?

- Screenshots are not favorable. It is needed to outputs the plots as figures.

- More in-depth discussion and comparison with alternative methods are needed. It is recommended to methods in Secure data transmission and trustworthiness judgement approaches against cyber-physical attacks in an integrated data-driven framework.

- FIgure 1 should be Table 1.

- Chapter 1 should be Section 1.

Round 2

Reviewer 1 Report

In the present form paper should be publised.

Reviewer 3 Report

Several previous comments were not addressed fully. 

comment #4, which part of the paper is based on implemented circuits rather than simulation? What results were obtained on real-world experiments?

comment #5, there are still several screenshots such as Figs. 24-29.

comment #6, there were no comparisons or discussions on the recommended article.

comment #7, Figure 20 should be a table.

The English writing needs major revision. For example, it is repetitive by writing "The conversion efficiency of the proposed topology is simulated by simulation experiments."

Round 3

Reviewer 3 Report

The paper has been revised.